# Difluoromethylornithine Induces Apoptosis through Regulation of AP-1 Signaling via JNK Phosphorylation in Epithelial Ovarian Cancer

**DOI:** 10.3390/ijms221910255

**Published:** 2021-09-23

**Authors:** Woo Yeon Hwang, Wook Ha Park, Dong Hoon Suh, Kidong Kim, Yong Beom Kim, Jae Hong No

**Affiliations:** 1Department of Obstetrics and Gynecology, Seoul National University Bundang Hospital, Seongnam 13620, Korea; wooyeonhwang@naver.com (W.Y.H.); summerwook@gmail.com (W.H.P.); sdhwcj@naver.com (D.H.S.); kidong.kim.md@gmail.com (K.K.); ybkimlh@snubh.org (Y.B.K.); 2Department of Obstetrics and Gynecology, Seoul National University College of Medicine, Seoul 03080, Korea

**Keywords:** AP-1, apoptosis, DFMO, JNK, ovarian cancer, polyamines

## Abstract

Difluoromethylornithine (DFMO), an irreversible inhibitor of ornithine decarboxylase (ODC), has promising activity against various cancers and a tolerable safety profile for long-term use as a chemopreventive agent. However, the anti-tumor effects of DFMO in ovarian cancer cells have not been entirely understood. Our study aimed to identify the effects and mechanism of DFMO in epithelial ovarian cancer cells using SKOV-3 cells. Treatment with DFMO resulted in a significantly reduced cell viability in a time- and dose-dependent manner. DFMO treatment inhibited the activity and downregulated the expression of ODC in ovarian cancer cells. The reduction in cell viability was reversed using polyamines, suggesting that polyamine depletion plays an important role in the anti-tumor activity of DFMO. Additionally, significant changes in Bcl-2, Bcl-xL, Bax protein levels, activation of caspase-3, and cleavage of poly (ADP-ribose) polymerase were observed, indicating the apoptotic effects of DFMO. We also found that the effect of DFMO was mediated by AP-1 through the activation of upstream JNK via phosphorylation. Moreover, DFMO enhanced the effect of cisplatin, thus showing a possibility of a synergistic effect in treatment. In conclusion, treatment with DFMO alone, or in combination with cisplatin, could be a promising treatment for ovarian cancer.

## 1. Introduction

Ovarian cancer is the most lethal gynecologic cancer and the seventh-most common cancer in women worldwide [1]. New treatments for ovarian cancer have been developed; however, these treatment strategies encounter a significant amount of resistance because of their heterogeneity in pathologic and genetic variants [2]. Therefore, studies on long-term maintenance therapy, or combinations of existing therapeutic approaches to overcome resistance and reduce mortality, are ongoing.

Cancer metabolism is an important component of cancer cell survival and an emerging target of anti-cancer agents. Polyamines, including putrescine, spermidine, and spermine, are essential for the survival of cancer cells [3]. An increase in intracellular polyamine levels is associated with increased cancer cell proliferation and tumorigenesis [4,5], whereas a decrease in intracellular polyamine levels caused by polyamine synthesis inhibitors is associated with cell growth inhibition in various cancers [6,7,8]. Therefore, inhibition of the polyamine pathway results in anti-tumor effects via apoptotic signaling pathways. 

The synthesis of polyamines is initiated through the activity of ornithine decarboxylase (ODC), which is the first rate-limiting enzyme in the synthesis of polyamines [9,10]. ODC is often upregulated in various cancer cells and contributes to cell proliferation and tumor growth by inducing an increase in the concentration of polyamines, including putrescine, spermidine, and spermine [11,12,13,14,15]. Increased ODC expression has been reported in various ovarian cancer cell lines, and the cancer cell line encyclopedia provides these data [16]. The anti-tumor effects of several ODC inhibitors have been investigated [17,18]. Difluoromethylornithine (DFMO) is one of the well-known inhibitors of ODC [3,19]. The inhibitory effects of DFMO have been explored in various cancers such as skin, breast, blood, prostate, and pancreatic cancers [20]. In addition, long-term administration of DFMO has shown tolerable safety profiles in chemopreventive clinical studies.

As an anti-cancer agent, DFMO exerts its inhibitory effect on cell proliferation and tumor growth through a process that involves complex relationships between ODC activity, polyamine levels, and oncogenes [3]. DFMO acts with several apoptosis-associated genes and proteins that have critical roles in regulating apoptosis. Such genes and proteins include caspases, Bcl-2 family proteins, FasL, cytochrome c, and PARP [21]. In addition, the AP-1 transcription factor, composed of Jun (c-Jun, JunB, and JunD) and Fos (c-Fos, FosB, Fra-1, and Fra-2) heterodimers [22], is mostly mediated via the JNK and p38MAPK pathways, which are important in regulating the transcription of a variety of genes involved in apoptosis [23]. 

Although DFMO is known to decrease polyamine levels through the irreversible inhibition of ODC, the exact mechanism by which it inhibits the proliferation of ovarian cancer cells has been rarely discussed. In an earlier experiment with a human ovarian cell line, DFMO was found to be a potential treatment agent for ovarian carcinoma [24,25]. However, to date, there have been no follow-up studies to examine the therapeutic effects of DFMO in ovarian cancer. In the present study, we investigated the effect and the cellular mechanism of DFMO in epithelial ovarian cancer cells with the purpose of using these data to establish a new promising strategy for the treatment of patients with epithelial ovarian cancer.

## 2. Results

### 2.1. DFMO Suppresses Cancer Survival and Induces Apoptotic Cell Death in Ovarian Cancer Cells

To study the effects of DFMO on the survival of human ovarian cancer cell lines, we performed cell viability and proliferation analyses. The half-maximal inhibitory concentration (IC50) values of DFMO in SKOV-3 and A2780 were determined using the PrestoBlue assay (Appendix A). Then, the SKOV-3 cells were treated with DFMO at different concentrations (0–100 µM) for 48 h, and then cell viability or proliferation was assessed using the PrestoBlue or CellTiter-Glo assays. DFMO significantly inhibited the viability of SKOV-3 cells in a dose-dependent manner (Figure 1A). Additionally, DFMO significantly reduced the proliferation of SKOV-3 cells (Figure 1B). To further confirm the induction of apoptosis by DFMO in human ovarian cancer cells, we performed an analysis of apoptosis activity and the apoptosis signaling pathway. The cells treated with DFMO were assessed using Annexin V or Caspase-3 assays, and the results showed that DFMO dramatically increased apoptosis and caspase-3 activity in a dose-dependent manner (Figure 1C,D). Moreover, DFMO increased the expression of proapoptotic protein Bax, cleaved caspase-3, and cleaved PARP, and suppressed the expression of ODC-1 and anti-apoptotic proteins Bcl-2 and Bcl-xL (Figure 1E). These results demonstrated that DFMO may inhibit survival and induce apoptotic cell death in ovarian cancer cells.

### 2.2. Polyamines Contribute to the Survival of Cancer Cells and Inhibit DFMO-Induced Cell Death in Ovarian Cancer Cells

To confirm the effect of polyamines on the growth and proliferation of human ovarian cancer cells, we performed cell viability and proliferation analyses of SKOV-3 cells. The cells were incubated with 50 μM spermidine or spermine for 1–5 days, and then cell viability and proliferation were determined using the PrestoBlue or CellTiter-Glo assays, respectively. As shown in Figure 2A, spermidine or spermine significantly increased the viability and proliferation of SKOV-3 cells in a time-dependent manner. To verify whether exogenous polyamines inhibited the apoptotic effect of DFMO and ODC-1 activity, we performed a cell viability analysis and an ODC-1 luciferase analysis of DFMO/polyamine-treated SKOV-3 cells. Spermidine and spermine significantly decreased the apoptotic effect of DFMO in SKOV-3 cells (Figure 2B). However, polyamines did not affect the DFMO-induced inhibition of ODC-1 activity (Figure 2C). The results revealed that polyamines are associated with cancer survival and DFMO-induced cell death but not with ODC-1 activity.

### 2.3. Combination of DFMO and Cisplatin Therapy Reduces Cancer Cell Survival and Promotes Apoptotic Cell Death in Ovarian Cancer Cells

To confirm the hypothesis that the addition of DFMO to a platinum-based anti-cancer drug would reduce cancer cell survival and induce apoptotic cell death in ovarian cancer cells, SKOV-3 cells were co-treated with 0–100 μM DFMO and 10 μM cisplatin for 48 h. The cells treated with a combination of 10 μM cisplatin/50 μM DFMO and 10 μM cisplatin/100 μM DFMO exhibited the largest decrease in cell viability (Figure 3A) and proliferation (Figure 3B), respectively, when compared with DFMO or cisplatin alone. In addition, the combination therapy increased apoptosis (Figure 3C, Appendix A) and caspase-3 activity (Figure 3D) in SKOV-3 cells. Moreover, there was a decrease in cell viability and proliferation when SKOV-3 cells were treated with two doses of cisplatin/DFMO (10 μM/100 μM) for 48 h, compared with when they were treated with DFMO or cisplatin monotherapy (Figure 3E,F). Western blot analysis showed that the combination therapy promoted the expression of pro- or anti-apoptotic genes in a dose-dependent manner in ovarian cancer cells when compared with DFMO or cisplatin monotherapy (Figure 3G). Collectively, these data revealed that the combination therapy of DFMO and cisplatin effectively induced apoptotic cell death in human ovarian cancer cells.

### 2.4. DFMO Induces Phosphorylation of JNK and Activates the AP-1 Signaling Pathway

To confirm whether the effects of DFMO were mediated via the AP-1 signaling pathway through JNK phosphorylation, we prepared a plasmid vector to measure AP-1 luciferase activity. The AP-1 luciferase vector was transiently transfected into the SKOV-3 cell, and the transfected cells were treated with DFMO for 48 h. We used the Luciferase Assay System (Promega) to evaluate the effect of DFMO on AP-1 activity in SKOV-3 cells. We found that AP-1 luciferase activity significantly increased when 50 µM and 100 µM of DFMO were used (Figure 4A). In further investigating the JNK and AP-1 signaling pathways with respect to DFMO use, Western blot analysis revealed that the expression levels of phospho-JNK, phospho-c-Jun, and AP-1 in SKOV-3 cells significantly increased after DFMO treatment (Figure 4B). The levels of expression of phospho-Bad and Bad proteins were also significantly regulated after DFMO treatment (Figure 4B). Furthermore, compared with DFMO monotherapy, a combination of cisplatin and DFMO promoted the induction of AP-1 luciferase activity even at a low concentration (10 μM cisplatin/1 μM DFMO) (Figure 4C). Moreover, Western blot analysis revealed that a combination of cisplatin and DFMO increased the activation of phospho-JNK and AP-1 signaling when compared with single-agent DFMO in SKOV-3 cells (Figure 4D). These results demonstrated that DFMO induced the activation of the AP-1 signaling pathway by triggering the phosphorylation of JNK in SKOV-3 cells.

### 2.5. AP-1 Mediates DFMO-Induced Apoptotic Cell Death in Ovarian Cancer Cells

To uncover the relationship between DFMO-induced apoptosis and AP-1 signaling, we used a specific AP-1 inhibitor, SR11302, to block AP-1 signaling, and analyzed the DFMO-induced apoptosis pathway in SKOV-3 cells. The Luciferase Assay revealed that SR11302 blocked the increase in AP-1 luciferase activity induced by DFMO in SKOV-3 cells (Figure 5A) and also inhibited DFMO-induced nucleus translocation of AP-1 in SKOV-3 cells (Figure 5B). Western blot analysis showed that SR11302 inhibited the expression of AP-1 through upstream phosphorylation of JNK and c-Jun, an AP-1 transcription factor subunit, induced by treatment with DFMO in SKOV-3 cells (Figure 5C). Furthermore, the expression of anti-apoptotic proteins, including Bcl-2 and Bcl-xL, was reduced by DFMO. Conversely, the expression of proapoptotic proteins Bad, Bax, cleaved caspase-3, and cleaved PARP significantly increased after DFMO treatment when compared with controls. These results provide additional evidence that AP-1 mediates DFMO-induced apoptotic cell death in SKOV-3 cells.

## 3. Discussion

This study found that ovarian cancer cells were likely to be dependent on polyamines and appeared to be sensitive to DFMO, which inhibits the polyamine pathway. In addition, exposure to DFMO alone, or in combination with cisplatin, reduced cell viability and proliferation in a dose-dependent manner. Previous studies have shown a reduction of cell viability and proliferation in cell lines, such as neuroblastoma and triple-negative breast cancer, upon treatment with DFMO alone [26,27,28]. However, DFMO showed a limited anti-tumor activity or limited inhibition of disease progression when used as a single agent in other cancers [29,30,31]. This may be due to the incomplete depletion of spermine, reverse conversion to spermidine and putrescine, or compensatory increased uptake of polyamines from the circulation or extracellular space. To overcome this limitation, DFMO has been examined in combination with various therapeutic methods. 

Combination with other chemopreventive agents, such as adenosine receptor inhibitors against breast cancer cells and polyamine transport inhibitors against murine squamous cell carcinoma, has enhanced the inhibition of growth and proliferation of tumor cells [32,33]. Another promising therapeutic approach found, which presented a new treatment target for glioblastoma, was the combined treatment of DFMO and radiation with tumor necrosis factor-related apoptosis-inducing ligand, an anti-cancer agent, or with temozolomide, an oral alkylating agent [34,35]. Several studies have also revealed that the combination of DFMO and nonsteroidal anti-inflammatory drugs is an effective and safe treatment option and decreases the risk of colon cancer [36].

Cisplatin is the cornerstone agent for ovarian cancer; however, its use is often restricted because of its toxic side effects [37]. Thus, we combined cisplatin with an effective and safe agent for the treatment approach. DFMO was approved by the US Food and Drug Administration for the treatment of African sleeping sickness [38]. DFMO is known as a particularly exceptional drug owing to its oral bioavailability and low toxicity for long-term administration [35]. Our study showed that combining DFMO and cisplatin significantly reduced the viability of ovarian cancer cells. We believe that this finding would provide an important rationale for further clinical trials related to maintenance therapy for ovarian cancer. 

The administration of DFMO with or without various conventional chemotherapy agents showed conflicting results in other clinical trials. In particular, clinical trials using DFMO for patients with anaplastic glioma revealed a survival benefit with or without procarbazine, lomustine, and vincristine (PCV) as adjuvant therapy [39]. However, in another study on glioblastoma multiforme, the addition of DFMO showed no advantage [40]. In another clinical trial, no differences were observed in the disease progression of prostate cancer patients treated with the combination of DFMO and doxorubicin or cyclophosphamide [41]. This may be related to the ODC levels of different tumors, as tumors with relatively low levels of ODC appear to respond better to DFMO [39,42]. In addition, chemotherapy may induce new genetic alterations in the initial tumor [3]. Therefore, the concurrent administration of DFMO and chemotherapy may theoretically have a distinct role in different cancer cells. 

We also investigated whether other important signaling pathways were involved in the mediation of DFMO-induced growth restriction and apoptosis. We observed that DFMO induced the phosphorylation of JNK and increased the expression of AP-1 in a dose-dependent manner. Consequently, AP-1 transcription factors were implicated in the process of cell apoptosis. This has been demonstrated in a study on a polyamine analog in various human breast cancer cell lines [43]. Polyamine analogs inhibited cell growth and induced apoptosis by modulating the expression of apoptotic proteins. The AP-1 family was one of the important proteins that mediated apoptosis. However, apoptosis activated by AP-1 was not observed in all breast cancer cell lines, suggesting that polyamine analog-induced apoptosis is associated with multiple apoptotic mechanisms in a cell-specific manner. Ovarian cancer has been considered as a highly heterogeneous disease in which distinct histological subtypes correlated with the differences in clinical outcomes [2]. This indicates that molecular targeted therapy, including DFMO, may be effective in some specific cell types, but not in all. We plan to extend our investigations to in vivo studies to support our results in the near future.

## 4. Materials and Methods

### 4.1. Cell Culture and Drugs

The SKOV-3 cell line was purchased from the American Type Culture Collection (Rockville, MD, USA). The cells were cultured in McCoy’s 5A (Welgene, Kyungsan, Korea) supplemented with 10% fetal bovine serum (FBS) and 1% penicillin-streptomycin (P/S) (Invitrogen, Carlsbad, CA, USA) in a humidified chamber with 5% CO_2_ at 37 °C. DFMO, cisplatin, spermine, spermidine, and SR11302 were purchased from Selleckchem (Houston, TX, USA) or Cayman Chemical (Ann Arbor, MI, USA) and were dissolved in dimethyl sulfoxide. The final concentrations of the culture media did not exceed 0.1%.

### 4.2. Cell Viability

Cell viability was measured using the PrestoBlue reagent (Invitrogen, CA, USA). SKOV-3 cells were seeded in 96-well plates (5 × 10^5^ cells/well) in McCoy’s 5A complete medium, containing 10% FBS and 1% P/S, and treated with 0–100 μM DFMO or with 10 μM cisplatin for 72 h. The treated cells were incubated with 10% PrestoBlue reagent for 60 min at 20–25 °C. The absorbance at 560 nm (Excitation) or 590 nm (Emission) was measured using an ELISA microplate reader (Molecular Devices, San Jose, CA, USA). All data were expressed as the percent of control.

### 4.3. Cell Proliferation

Cell proliferation was assessed using the CellTiter-Glo assay kit (Promega, Madison, WI, USA). Cells were seeded in 96-well plates in McCoy’s 5A complete medium, containing 10% FBS and 1% P/S, and treated with 0–100 μM DFMO or with 10 μM cisplatin for 72 h. The CellTiter-Glo reagent was added to the treated cells with a volume equal to that of the medium in each well. The cells were incubated at room temperature for 10 min, and luminescence was measured using a luminescence plate reader (Berthold, Germany).

### 4.4. Caspase 3/7 Activity

SKOV-3 cells were seeded in a white-walled 96-well plate (1 × 10^5^ cells/well), cultured in McCoy’s 5A complete medium for 24 h, and treated with 0–100 μM DFMO or with 10 μM cisplatin for 72 h. The treated cells were harvested and incubated with 100 μL of Caspase-Glo 3/7 Reagent for 30 min at room temperature. Luminescence of each sample was measured using the using an LMAX II 384 luminometer (Molecular Devices, Sunnyvale, CA, USA). All data were expressed as the fold induction of control.

### 4.5. Ornithine Decarboxylase-1 (ODC-1) Activity

The promoter of the human ornithine decarboxylase-1 gene was PCR-amplified from genomic DNA and cloned into the pGL-4 basic reporter vector (Promega). The primer sequence is as follows: forward 5′- aagctagcAAAAAGTGTCCCCAATTCAAGTGCAGTGCC -3′, reverse 5′- actcgagCGAAAATAAAAACTGGAAGGAAACTGAAGGCG -3′. The pODC1-Luc vectors were transiently transfected into SKOV-3 cells using the FuGENE HD transfection reagent (Promega) according to the manufacturer’s instructions. The pODC1-Luc/SKOV-3 cells were seeded and treated with 50 or 100 μM DFMO for 48 h in a 96-well plate. Then, the cells were treated with 10 or 50 μM polyamine for 48 h, and the activity of firefly luciferase was measured using the Luciferase Assay kit (Promega) according to the manufacturer’s instructions.

### 4.6. AP-1 Luciferase Activity

The luciferase vector contained the AP-1 response element, which was purchased from Promega. The reporter vectors were transiently transfected into SKOV-3 cells using the FuGENE HD transfection reagent (Promega). The transfected cells were seeded into a 96-well plate for 24 h and treated with 0–100 μM DFMO or 10 μM cisplatin for 48 h. The activity of AP-1 luciferase was determined using the Luciferase Assay System (Promega).

### 4.7. Annexin V-FITC/PI Assay via Flow Cytometry

To analyze the apoptotic cell death rate, an Annexin V-FITC assay was performed using the FITC Annexin V Apoptosis Detection Kit (BD Bioscience, San Diego, CA, USA) according to the manufacturer’s protocols. The cells were seeded in a 6-well plate (5 × 10^5^ cells/well) and treated with 0–100 μM DFMO or with 10 μM cisplatin for 72 h. After treatment, the supernatant and cells were harvested and centrifuged at 1500 rpm for 7 min. The cell pellets were then resuspended in 100 μL of 1× Annexin Binding Buffer. FITC Annexin V (5 μL) and PI (1 μg/mL) were added. Then, the cells were incubated in the dark for 15 min at room temperature (25 °C). After incubation, 400 μL of 1× Binding Buffer was added to each sample. The samples were analyzed using flow cytometry within 1 h. Flow cytometry was performed using a FACSCalibur flow cytometer (BD Bioscience, CA, USA) by analyzing at least 10,000 cells per sample. Results were presented as percentage (%) of the total gated number of cells.

### 4.8. Subcellular Fractionation

Subcellular organelle fractions from SKOV-3 cells were prepared through differential centrifugation. The cells were harvested in phosphate-buffered saline (PBS). Cell pellets were resuspended in each sub-organelle fractionation buffer and homogenized using a Dounce homogenizer. The cell lysates in the nuclei preparation buffer (NB) (1.5 mM MgCl2, 10 mM HEPES–KOH (pH 7.9), 0.5 mM DTT, 10 mM KCl, and 0.5 mM phenylmethylsulfonyl fluoride) were centrifuged at 2700 g for 10 min at 4 °C. The cell pellets re-homogenized in NB containing 0.2% NP-40 were centrifuged at 200 g for 5 min to prepare the nuclear fraction (Nu). Then, the cellular and nuclear pellets were resuspended in a lysis buffer (PBS, pH 7.4, 2% SDS, 1 mM PMSF, 1 g/mL aprotinin, 1 g/mL pepstatin A, and 2 g/mL leupeptin) for Western blot analysis. The protein concentration was determined using the BCA method (Pierce, Rockford, IL, USA).

### 4.9. Western Blot

Protein lysate was prepared in PBS (pH 7.4, 2% SDS, 1 mM phenylmethylsulfonyl fluoride, 1 g/mL aprotinin, 1 g/mL pepstatin A, and 2 g/mL leupeptin), separated using a 12% or 15% SDS–PAGE, and transferred to polyvinylidene difluoride membranes. Blots were probed with the following primary antibodies (1:1000 or 1:500 dilute factor): ODC-1(ab66067), phospho-JNK^Thr183/Tyr185)^(9255S) phospho-c-Jun ^Ser63^(91952S), AP-1(MBS9412084), phospho-Bad^Ser112^(5284S), HDAC1(34589S), Bcl-2(ab182858), Cleaved PARP (ab32064), Bax(ab32503), Cleaved Caspase-3(ab32042), JNK(9252S), Bad(9239S), c-Fos(2250S), Histone H3(4499S), GAPHD(5174S), and Hsp90(4877S). Equivalent protein loading was verified by blotting with anti-β-actin antibodies (4970). All blots were visualized using an enhanced Immobilon ECL substrate (Merck Millipore, Billerica, MA, USA).

### 4.10. Statistical Analysis

Data were presented as mean ± standard deviation. All analyzed data were graphed using GraphPad Software (San Diego, CA, USA). Student’s t-test was used to compare the different groups. A *p*-value < 0.05 was considered statistically significant.

## 5. Conclusions

As there is a persistent demand to develop more efficient anti-ovarian-cancer drugs with less toxicity, the effect of DFMO in different subtypes of ovarian cancer cells and its synergistic effects with other anti-tumor agents should be further explored. The results of the present study showed that DFMO, a specific inhibitor of ODC, induced apoptosis by increasing the expression of AP-1 and JNK phosphorylation in ovarian cancer cells either alone or in combination with cisplatin. Therefore, our study results provided a rational basis for DFMO as a novel therapeutic option for patients with ovarian cancer. 

## Figures and Tables

**Figure 1 ijms-22-10255-f001:**
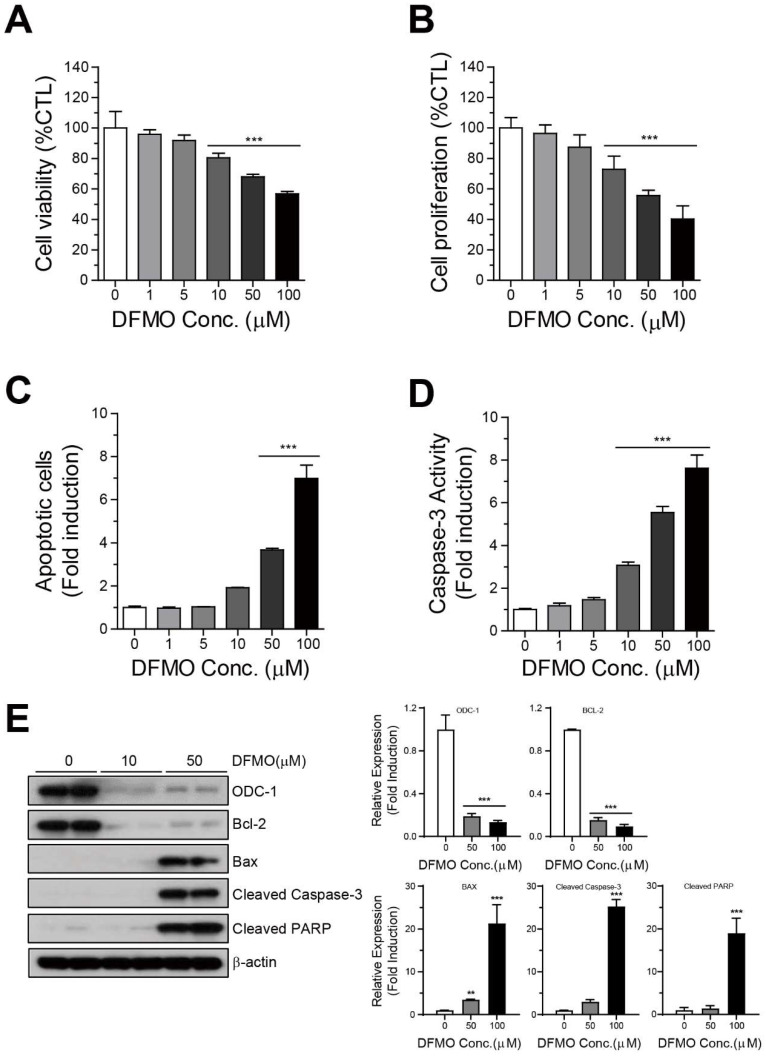
Inhibition of cell survival and induction of apoptotic cell death by DFMO in SKOV-3 cells. (**A**) SKOV-3 cells were cultured with varying concentrations of DFMO (from 0 to 100 μM) for 48 h, and cell viability was detected using the PrestoBlue assay. (**B**) Cell proliferation was determined by analyzing ATP content in DFMO-treated SKOV-3 cells using the CellTiter-Glo assay. (**C**) Flow cytometric analysis of Annexin V-FITC/PI-stained SKOV-3 cells treated with the control (0.1% DMSO) or 0–100 μM DFMO was used to determine the apoptotic rates of SKOV-3 cells under various concentrations of DFMO. (**D**) Caspase-3 activity in DFMO-treated SKOV-3 cells was measured using the Luciferase Assay (Caspase-Glo 3/7 Assay system). (**E**) Western blot and quantification analysis for the apoptotic proteins Bcl-2, Bcl-xL, Bax, cleaved caspase-3, and cleaved PARP in cells treated with DFMO. The controls were treated with 0.1% DMSO. Data are expressed as mean ± SD. ** *p* < 0.01, *** *p* < 0.001 compared with the controls.

**Figure 2 ijms-22-10255-f002:**
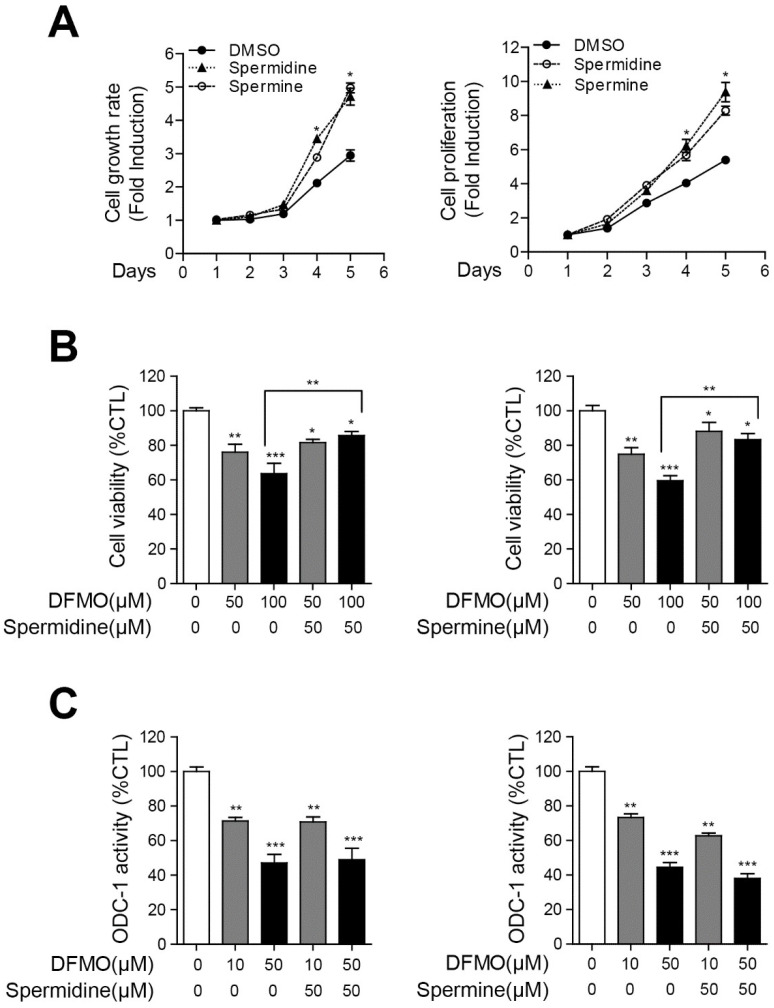
Interruption in DFMO-induced cell death by polyamines in SKOV-3 cells. (**A**) SKOV-3 cells were incubated with polyamines for 1–5 days. Cell growth rate and proliferation were determined using the PrestoBlue and CellTiter-Glo assays, respectively. (**B**) Polyamines were treated with a concentration of 0 or 50 μM for 48 h in DFMO-treated SKOV-3 cells, and the cell viability in DFMO/polyamines-treated SKOV-3 cells was measured using the PrestoBlue assay. (**C**) ODC-1 activity in DFMO/polyamines-treated SKOV-3 cells was detected using the Luciferase Assay (Luciferase Assay System). The controls were treated with 0.1% DMSO. Data are expressed as mean ± SD. * *p* < 0.05, ** *p* < 0.01, *** *p* < 0.001 compared with the controls.

**Figure 3 ijms-22-10255-f003:**
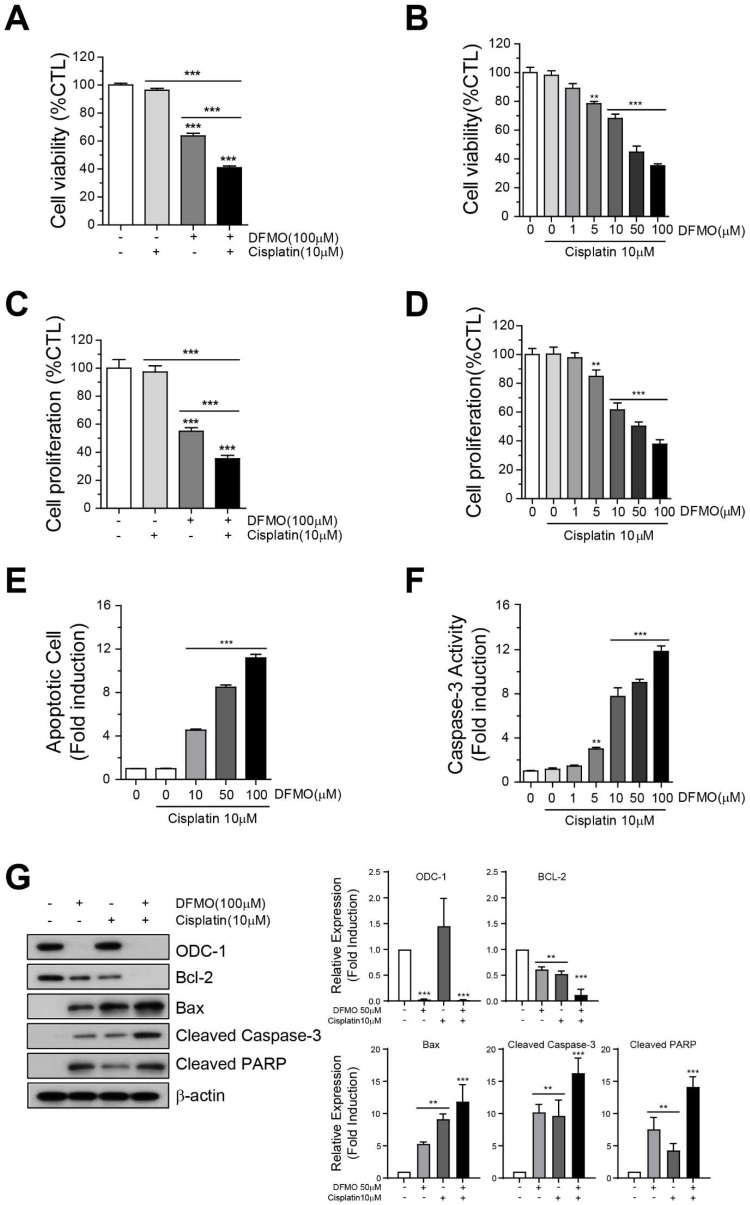
Effect of combination therapy using DFMO and cisplatin on SKOV-3 cell survival and apoptotic cell death. Cell viability was determined of cells treated with (**A**) 10 μM cisplatin, 100 μM DFMO, or (**B**) 10 μM cisplatin/0–100 μM DFMO for 48 h using the PrestoBlue assay and CellTiter-Glo assay. (**C**) Proliferation was determined in cells treated with (**C**) 10 μM cisplatin, 100 μM DFMO, or (**D**) 10 μM cisplatin/0–100 μM DFMO for 48 h. (**E**) Flow cytometric analysis of Annexin V-FITC/PI-stained SKOV-3 cells treated with 10 μM cisplatin/0–100 μM DFMO for 48 h was performed to determine the apoptotic rates of SKOV-3 cells under various concentrations of DFMO. (**F**) Caspase-3 activity in cisplatin/DFMO-treated SKOV-3 cells was measured using the Luciferase Assay (Caspase-Glo 3/7 Assay system). (**G**) Western blot and quantification analysis for the apoptotic proteins Bcl-2, Bcl-xL, Bax, cleaved caspase-3, and cleaved PARP in cells treated with DFMO. The controls were treated with 0.1% DMSO. Data are expressed as mean ± SD. ** *p* < 0.01, *** *p* < 0.001 compared with the controls.

**Figure 4 ijms-22-10255-f004:**
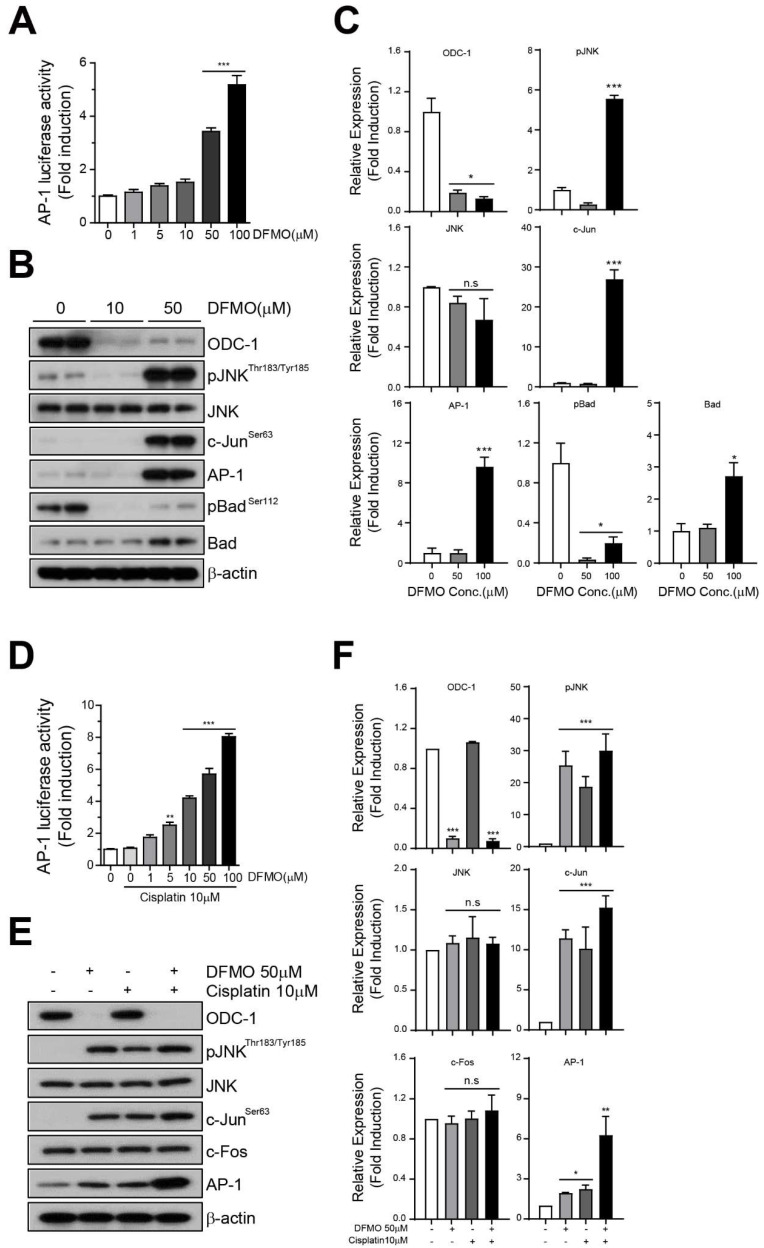
Induction of AP-1 activity and JNK signaling using the combination therapy of DFMO and cisplatin. An AP-1 luciferase plasmid vector was transiently transfected into SKOV-3 cells for 48 h. The transfected cells were seeded in cell culture plates and treated with cisplatin or DFMO for 48 h. The AP-1 luciferase activity of DFMO- or cisplatin/DFMO-treated cells was measured using the Luciferase Assay System (**A**,**D**). Western blot and quantification of the proteins of JNK signaling (phospho-JNK, JNK, phospho-cJun, AP-1, c-Fos), and apoptosis (phospho-Bad, Bad) in DFMO- or cisplatin/DFMO-treated cells (**B**,**C**,**E**,**F**). The controls were treated with 0.1% DMSO. Data are expressed as mean ± SD. * *p* < 0.05, ** *p* < 0.01, *** *p* < 0.001 compared with the controls.

**Figure 5 ijms-22-10255-f005:**
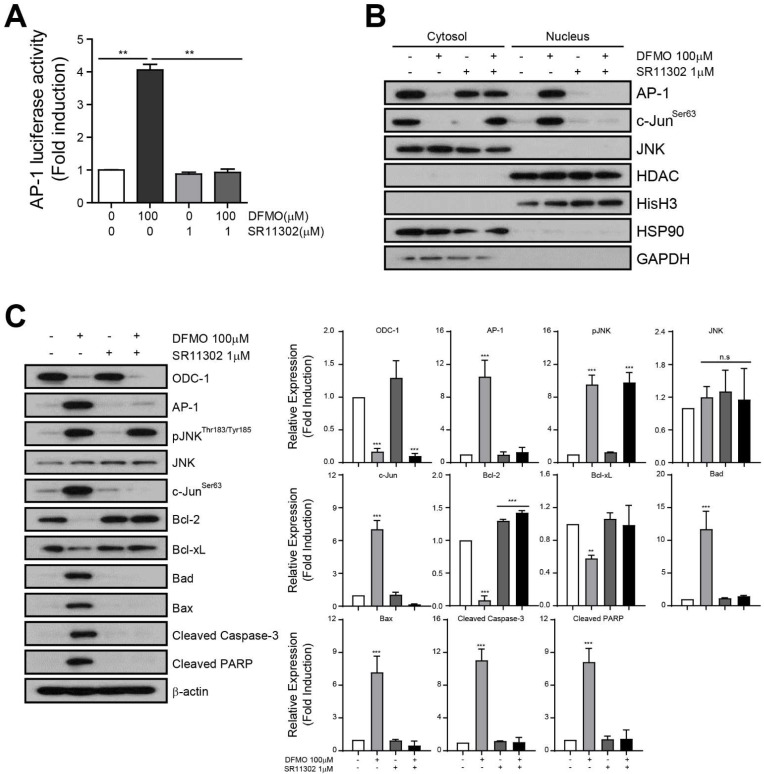
Induction of DFMO-induced apoptotic cell death through the regulation of AP-1 activity. The transfected cells were seeded in cell culture plates and treated with DFMO, SR11302, or DFMO/SR11302 for 48 h. (**A**) The AP-1 luciferase activity in the treated cells was determined using the Luciferase Assay System. (**B**) Western blot of organelle extracts (10 μg) was performed using the following markers: nucleus; HDAC, Histone H3/ cytosol: Hsp90, GAPDH. (**C**) Western blot analysis of the JNK signaling pathway detected JNK signaling proteins (phospho-JNK, JNK, phospho-cJun, AP-1), anti-apoptotic proteins (Bcl-2, Bcl-xL), and proapoptotic proteins (Bad, Bax cleaved caspase-3, cleaved PARP) in DFMO-, SR11302-, or DFMO/SR11302-treated cells, respectively. The controls were treated with 0.1% DMSO. Data are expressed as mean ± SD. * *p* < 0.05, ** *p* < 0.01, *** *p* < 0.001 compared with the controls.

## Data Availability

The data presented in this study are available upon request from the corresponding author.

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
