# Peer review of "Difluoromethylornithine Induces Apoptosis through Regulation of AP-1 Signaling via JNK Phosphorylation in Epithelial Ovarian Cancer"

_ijms, 2021, doi:10.3390/ijms221910255_

Round 1

Reviewer 1 Report

  1. To avoid single cell line bias and ensure comprehensive results across more genetic backgrounds, the authors have to include results on at least a second cell line for Fig-1, -3, as they did for the rest of the manuscript with two ovarian cancer cell lines, only cell line is not be accepted
  2. Need to test the cytotoxicity of normal ovarian cells, and provide IC50 value of normal and cancer cells
  3. Apoptosis was association with mitochondrial membrane depolarization, the author should be test the DFMO whether affect the mitochondrial dysfunction in ovarian cancer cell line
  4. In Fig-3A~3D, need provide the DFMO-treated alone cells (100 uM), it should be compared with DFMO combined with cisplatin treated-cells.
  5. In Fig-3G: from the results of Fig-1E, western blot analysis found the increased the expression of Bax and cleaved PARP in DFMO (50 uM) treated-cells, but why the DFMO (100 uM) treated-cells not induce the Bax and cleaved PARP expression in Fig-3G (line 2), That is not logic, it need examine it.
  6. In Fig-4B and 4D section: Fig-4B: the author demonstrated that not increased the expression of p-JNK,p-cJun and AP-1 in DFMO (50 uM) treated-cells (line 3,4), the same question, why the DFMO (50 uM) treated-cells increased the p-JNK,p-cJun and AP-1 in DFMO (50 uM) treated-cells in Fig-4D (line 2), it need examine it. That is not logic.
  7. In Fig-5, the author should be provide the cell viability and cell death assay in treated with DFMO, SR11302, or DFMO/SR11302 cells.
  8. Need the animal model to strong support this in vitro study.
  9. They have analyzed, by western Blot, the cell content of several signaling proteins related to cell apoptosis. They showed representative blots (of three independent experiments as informed in the figure legends), however the quantitative analyses by densitometry of the protein bands are lacking. This quantitative analysis will certainly improve the results and must be presented graphically.
  10. All antibody need provide the product number and dilute factor in western blotting
  11. There are no conclusions from the conducted research. What application in clinical practice may have these results?

Author Response

Thank you very much for the review of our manuscript (IJMS-1368450). The comments of the review were constructive and have been used to revise and improve the manuscript. We enclose a revised version of our paper titled “Difluoromethylornithine Induces Apoptosis through Regula-tion of AP-1 Signaling via JNK Phosphorylation in Epithelial Ovarian Cancer” by Woo Yeon Hwang, Wook Ha Park, Dong Hoon Suh, Kidong Kim, Yong Beom Kim, Jae Hong No as a submission to International Journal of Molecular Science. We marked up with the “Track changes” function to make our revisions. The following is an itemized account of the changes in the manuscript made in response to the comments.

Point 1: To avoid single cell line bias and ensure comprehensive results across more genetic backgrounds, the authors have to include results on at least a second cell line for Fig-1, -3, as they did for the rest of the manuscript with two ovarian cancer cell lines, only cell line is not be accepted

Response 1: Thank you for providing valuable comment. We have experimented in other cell line previously but we didn’t do the same experiment in this study. We only have cell viability data. We have attached in a new supplement to the article as supplementary Figure S1. We couldn’t do the additional experiment because we only gave 7days for revision, but if you give us more time, we are willing to perform further experiments.

Modified text: In line 85-88 and in 143-147 we have added the text about supplementary Figure S1.

In line 85-88

Also, we performed cell viability analyses of A2780 and OVCAR8 cells with 0–100 μM DFMO previosly. DFMO significantly inhibited the viability of both cells in a dose-dependent manner (Supplementary Figure S1).

In 143-147

Also, we performed cell viability analyses of A2780 and OVCAR8 cells with 0–100 μM DFMO and 4 μM cisplatin for 48 h previously. The cells treated with a combination of 2 μM cisplatin/100 μM DFMO and 4 μM cisplatin/100 μM DFMO exhibited the largest decrease in cell viability (Supplementary Figure S1).

Point 2: Need to test the cytotoxicity of normal ovarian cells, and provide IC50 value of normal and cancer cells

Response 2: Thank you for your valuable comment. It is an important point and we agree your opinion, but normal cancer cell is commercially unavailable and there is a difficulty in making and maintaining cell line. Therefore, we used the concentration that showed a decrease in cell viability and proliferation to concentration that began to decrease by about 50%.

Point 3: Apoptosis was association with mitochondrial membrane depolarization, the author should be test the DFMO whether affect the mitochondrial dysfunction in ovarian cancer cell line

Response 3: We appreciate your valuable comment. Metabolically active cells produce ATP as energy for respiration and other vital processes, and the major function of mitochondria is a regulation of energy metabolism and generation of ATP in the cell. So, the mitochondria is called that the power of house. We performed the cell proliferation assay in DFMO-treated ovarian cancer cell. The value of cell proliferation was measured by the ATP contents in DFMO-treated SKOV-3 cell using Cell-Titer Glo assay system. In the Fig 1B, the ATP contents could be considered mitochondrial functional activity. However, if you think you need to identify it through additional experiments, we’ll determine the mitochondrial activity such as ROS production, MTT activity, mitochondrial membrane potential or OCR (oxygen consumption rate).

Point 4: In Fig-3A~3D, need provide the DFMO-treated alone cells (100 uM), it should be compared with DFMO combined with cisplatin treated-cells.

Response 4: Thank you for your valuable comment. We have compared DFMO-treated alone cells (100 uM) with cisplatin 10 uM and DFMO combined with cisplatin in figure A and C. (Previsous version Figure E and F). We have re-ordered the figure to show the results clearly.

Modified text:

Figure 3. Effect of combination therapy using DFMO and cisplatin on SKOV-3 cell survival and apoptotic cell death. Cell viability was determined of cells treated with (A) 10 μM cisplatin, 100 μM DFMO or (B) 10 μM cisplatin/0–100 μM DFMO for 48 h using the PrestoBlue assay and CellTiter-Glo assay. (C) proliferation was determined of cells treated with (C) 10 μM cisplatin, 100 μM DFMO, or (D) 10 μM cisplatin/0–100 μM DFMO for 48 h. (E) Flow cytometric analysis of Annexin V‑FITC/PI‑stained SKOV-3 cells treated with 10 μM cisplatin/0–100 μM DFMO for 48 h was performed to determine the apoptotic rates of SKOV-3 cells under various concentrations of DFMO. (F) Caspase-3 activity in cisplatin/DFMO-treated SKOV-3 cells was measured using the Luciferase Assay (Caspase-Glo 3/7 Assay system). (G) Western blot analysis detected the apoptotic proteins Bcl‑2, Bcl-xL, Bax, cleaved caspase‑3, and cleaved PARP in cells treated with DFMO. The controls were treated with 0.1% DMSO. Data are expressed as mean ± SD. *P < 0.05, **P < 0.01, ***P < 0.001 compared with the controls.

Point 5: In Fig-3G: from the results of Fig-1E, western blot analysis found the increased the expression of Bax and cleaved PARP in DFMO (50 uM) treated-cells, but why the DFMO (100 uM) treated-cells not induce the Bax and cleaved PARP expression in Fig-3G (line 2), That is not logic, it need examine it.

Response 5: Thank you for your valuable comment. We are so sorry that we made such a big mistake. We reviewed the research data again and there was a mistake in converting the experimental results into figures. We attached the figure again.

Point 6: In Fig-4B and 4D section: Fig-4B: the author demonstrated that not increased the expression of p-JNK,p-cJun and AP-1 in DFMO (50 uM) treated-cells (line 3,4), the same question, why the DFMO (50 uM) treated-cells increased the p-JNK,p-cJun and AP-1 in DFMO (50 uM) treated-cells in Fig-4D (line 2), it need examine it. That is not logic.

Response 6: Thank you for your valuable comment. Once again we are so sorry that we made such a big mistake. The concentration of DFMO in Fig B was 0, 10, 50 instead of 0, 50, 100.

Point 7: In Fig-5, the author should be provide the cell viability and cell death assay in treated with DFMO, SR11302, or DFMO/SR11302 cells.

Response 7: We appreciate your valuable suggestion. We provided the cell viability and cell death assay for DFMO in Figure 1, but we didn’t provide it for SR11302 or DFMO/SR11302. In our study, we wanted to evaluate whether DFMO induce the activation of the AP-1 signaling pathway and used SR11302, a specific AP-1 inhibitor, to unconver the relationship between DFMO-induced apoptosis and AP-1 signaling. We only wanted to know if DFMO is involved in the AP-1 signaling, not the effect of AP-1 inhibitor on cell viability and cell death. Therefore, we thought that the cell viability and cell death assay in treated with SR11302 or DFMO/SR11302 cells were unnessesary.

Point 8: Need the animal model to strong support this in vitro study.

Response 8: We agree with your suggestion. We are planning to extend our investigations in the near future using animal model. We added this in manuscript in line 301-302.

Modified text: We plan to extend our investigations to in vivo study to support our results in near future.

Point 9: They have analyzed, by western Blot, the cell content of several signaling proteins related to cell apoptosis. They showed representative blots (of three independent experiments as informed in the figure legends), however the quantitative analyses by densitometry of the protein bands are lacking. This quantitative analysis will certainly improve the results and must be presented graphically.

Response 9: We agree with your suggestion. If you think this is necessary, we’ll analyse the density of protein bands using Image J software.

Point 10: All antibody needs provide the product number and dilute factor in western blotting

Response 10: We agree your comment for information of antibodies. And we explained the product number and dilution factor in western blot section in line 388-395.

Modified text: ODC-1(ab66067), phospho-JNKThr183/Tyr185)(9255S) phospho-c-Jun Ser63(91952S), AP-1(MBS9412084), phospho-BadSer112(5284S), HDAC1(34589S), Bcl-2(ab182858), Cleaved PARP (ab32064) , Bax(ab32503), Cleaved Caspase-3(ab32042), JNK(9252S), Bad(9239S), c-Fos(2250S), Histone H3(4499S), GAPHD(5174S) and Hsp90 (4877S). Equivalent protein loading was verified by blotting with anti-β-actin antibodies (4970). All blots were visualized using an enhanced Immobilon ECL substrate (Merck Millipore).

Point 11: There are no conclusions from the conducted research. What application in clinical practice may have these results?

Response 11: Thank you for your valuable comment. We added the conclusion regarding application in clinical practice of our results in conclusion section in line 406-407.

Modified text: Therefore, our study results provided a rational basis for DFMO as a novel therapeutic option for patients with ovarian cancer.

Reviewer 2 Report

Authors present a well-conducted study about the effect of DFMO on ovarian cancer cell proliferation and apoptosis.

Let me ask the following questions:

Fig 1a and b: Effects of DFMO seem to be linear, regarding dose-dependency. In higher doses, do authors expect a further linear course, or rather a flattening of the curve?

In the introduction, authors reveal the role of FasL, cytochrome c, PARP, and others in apoptosis, but they did not measure them. Would measurements of these genes and proteins be of value in the further evaluation of CFMO-induced apoptotic pathways?

Discussion: Authors speculate about the dependency of intracellular ODC activity and DFMO efficiency. Wouldn´t it then be good to compare at least ODC mRNA levels between some ovarian cancer cell lines in advance?

Figure 3C: flow cytometry of Annexin V stained cells- can authors show the flow cytometry plots here? This would make depiction clearer and easier to understand. And can we also see some ICC staining images of Annexin and Caspase-3?

4.1 SKOV-3 cell line is know to have a resistance against Cisplatin. In what way may this resistence have influenced the results after Cisplatin application? Please discuss.

4.6. AP-1 response element transfection: I know from my own experience that transfection efficiency of FUGENE HD transfection is very low. Can authors please report their transfection efficiency?

Author Response

Thank you very much for the review of our manuscript (IJMS-1368450). The comments of the review were constructive and have been used to revise and improve the manuscript. We enclose a revised version of our paper titled “Difluoromethylornithine Induces Apoptosis through Regula-tion of AP-1 Signaling via JNK Phosphorylation in Epithelial Ovarian Cancer” by Woo Yeon Hwang, Wook Ha Park, Dong Hoon Suh, Kidong Kim, Yong Beom Kim, Jae Hong No as a submission to International Journal of Molecular Science. We marked up with the “Track changes” function to make our revisions. The following is an itemized account of the changes in the manuscript made in response to the comments.

Point 1: Fig 1a and b: Effects of DFMO seem to be linear, regarding dose-dependency. In higher doses, do authors expect a further linear course, or rather a flattening of the curve?

Response 1: We didn’t experiment for higher doses as it can induce cell death in normal cells which we don’t want. However, we expect a further linear course for higher doses.

Point 2: In the introduction, authors reveal the role of FasL, cytochrome c, PARP, and others in apoptosis, but they did not measure them. Would measurements of these genes and proteins be of value in the further evaluation of CFMO-induced apoptotic pathways?

Response 2: Thank you for your comment. I agree with your opinion. I think that it is necessary to determine the expression level of Fas-L, cytochrome c, PARP or other genes. But, in the study, we confirmed the overall mechanisms of apoptotic cell death through the change of cell proliferation, caspase-3 activity or apoptosis molecules. So, we decided not to identify the expression of proteins. However, if you think you need to identify it through additional experiments, we’ll determine the expression of FasL, cytochrome c, PARP, and others gene or protein using immunoblot.

Point 3: Discussion: Authors speculate about the dependency of intracellular ODC activity and DFMO efficiency. Wouldn´t it then be good to compare at least ODC mRNA levels between some ovarian cancer cell lines in advance?

Response 3: We appreciate your suggestion. We agree that it would be good to compare at least ODC mRNA level in ovarian cancer cell lines. We obtained the data regarding the ODC1 RNAseq gene expression of ovarian cancer cell lines from Cancer Cell Line Encyclopedia (CCLE) as follows. We added this information in line 52-54.

ODC1 RNAseq gene expression level of ovarian cancer cell lines

Cell line

ODC1 level

59M

80.3

A2780

80.4

CAOV3

32.2

CAOV4

59.2

COV318

48.0

COV362

103.2

COV434

71.7

COV644

185.0

EFO21

85.5

EFO27

68.0

ES2

174.0

FUOV1

101.8

HEYA8

151.4

IGROV1

162.0

JHOC5

53.2

JHOM1

26.7

JHOM2B

66.2

JHOS2

48.3

JHOS4

25.5

KURAMOCHI

63.0

MCAS

246.3

NIHOVCAR3

100.0

OAW28

37.0

OAW42

139.5

OC314

92.1

OELE

48.9

ONCODG1

121.2

OV56

7.0

OV7

101.7

OV90

119.3

OVCAR4

33.5

OVCAR8

86.5

OVISE

32.5

OVK18

121.6

OVKATE

25.1

OVMANA

151.1

OVSAHO

60.3

OVTOKO

9.7

RMGI

87.8

RMUGS

21.2

SKOV3

24.3

SNU119

64.4

SNU840

29.6

SNU8

68.2

TOV112D

66.9

TOV21G

35.3

TYKNU

205.0

Modified Text: Increased ODC expression has been reported in various ovarian cancer cell lines and the cancer cell line encyclopedia provide these data.

Point 4: Figure 3C: flow cytometry of Annexin V stained cells- can authors show the flow cytometry plots here? This would make depiction clearer and easier to understand. And can we also see some ICC staining images of Annexin and Caspase-3?

Response 4: In the Fig.3C and 3D data, we determined the apoptotic cell death in DFMO-treated ovarian cancer cell. To confirm DFMO-induced cell death, we were performed the Annexin V/PI detection assay by flow cytometry and Caspase-Glo3/7 Assay by luminometer. The Annexin V/PI assay was determined the fluorescence of live or dead cell, and Caspase-Glo3/7 Assay was measured the luminescence in the cells. So, we didn’t have the ICC staining images of Annexin and Caspase-3. However, if you think you need to identify it through additional experiments, we’ll perform the ICC staining for Annexin and Caspase-3. We added the flow cytometry plots in supplementary Figure 2.

Point 5: 4.1 SKOV-3 cell line is known to have a resistance against Cisplatin. In what way may this resistence have influenced the results after Cisplatin application? Please discuss.

Response 5: Thank you for your valuable comment. As you said SKOV-3 celll line is known to have a resistance against Cisplatin. Resistance may have influence to rate of cell growth or proliferation. However, previous reports and our study show that cisplatin induces apoptosis in SKOV-3 cell. It is also known that cisplatin activates multiple signaling pathways regarding apoptosis. Ovarian cancer is a very heterogeneous disease and tumor heterogeneity play an important role in treatment failure. DFMO combined with cisplatin may be a novel treatment option for ovarian cancer even in cisplatin resistance ovarian cancer in future with further studies.

Point 6: 4.6. AP-1 response element transfection: I know from my own experience that transfection efficiency of FUGENE HD transfection is very low. Can authors please report their transfection efficiency?

Response 6: We have several experiences in FUGENE HD transfection, therefore it is prepared.  We couldn’t report the exact rate of transfecton efficiency, bur fortunately, transfection was successful and we were able to earn results as shown in figure 4.

Round 2

Reviewer 1 Report

  1. It is described that this study was undertaken to examine anticancer effect of DFMO using SKOV-3 cells and that DFMO exhibited strong anticancer effect based on induction of apoptosis in SKOV-3 cells. However, primary problem of this paper is no evidence for the selective anticancer effect of DFMO, because it is not clear whether cytotoxic effect of DFMO on SKOV-3 cells was due to general cytotoxicity or due to specific anticancer activity, as the cytotoxicity was not compared between SKOV-3 cells and normal reference cells. In addition, the author must provide the IC50 of DFMO in SKOV-3 and other ovarian cancer cells.
  2. In all western blotting, the author should be providing the quantitative analysis will certainly improve the results and must be presented graphically.

Author Response

Thank you very much again for the review of our manuscript (IJMS-1368450). The comments of the review were constructive and have been used to revise and improve the manuscript. We enclose a revised version of our paper titled “Difluoromethylornithine Induces Apoptosis through Regula-tion of AP-1 Signaling via JNK Phosphorylation in Epithelial Ovarian Cancer” by Woo Yeon Hwang, Wook Ha Park, Dong Hoon Suh, Kidong Kim, Yong Beom Kim, Jae Hong No as a submission to International Journal of Molecular Science. We marked up with the “Track changes” function to make our revisions. The following is an itemized account of the changes in the manuscript made in response to the comments.

Point 1: It is described that this study was undertaken to examine anticancer effect of DFMO using SKOV-3 cells and that DFMO exhibited strong anticancer effect based on induction of apoptosis in SKOV-3 cells. However, primary problem of this paper is no evidence for the selective anticancer effect of DFMO, because it is not clear whether cytotoxic effect of DFMO on SKOV-3 cells was due to general cytotoxicity or due to specific anticancer activity, as the cytotoxicity was not compared between SKOV-3 cells and normal reference cells. In addition, the author must provide the IC50 of DFMO in SKOV-3 and other ovarian cancer cells.

Response 1: Thank you for providing valuable comment. It is an important point and we agree your opinion, but unfortunately we didn’t compared cytotoxicity between SKOV-3 cells and normal reference cells. However, we provided the IC50 of DFMO in SKOV-3 and A2780 in Supplement figure 1. If you think the cytotoxicity test compared between SKOV-3 cells and normal reference cells, we are willing to perform further experiments.

Modified text:

Supplemental Figure 1. IC50 value for DFMO in ovarian cancer cells. (A) SKOV-3 and (B) A2780 were cultured with various concentrations of DFMO 1nM - 500μM for 48h, and the cell viability was determined using the PrestoBlue assay. All values are averages of replicates expressed relative to cell viability values in untreated cells normalized to 100%. Cell viability curves represent n = 4 experiments with 3 replicates per drug concentration for each experiment.

Point 2: In all western blotting, the author should be providing the quantitative analysis will certainly improve the results and must be presented graphically.

Response 2: Thank you for your valuable comment. We provided the quantitative analysis presented graphically in each figure next to western blot analysis.

Modified text:

Figure 1. Inhibition of cell survival and induction of apoptotic cell death by DFMO in SKOV-3 cells. (A) SKOV-3 cells were cultured with varying concentrations of DFMO (from 0 to 100 μM) for 48 h, and cell viability was detected using the PrestoBlue assay. (B) Cell proliferation was determined by analyzing ATP content in DFMO-treated SKOV-3 cells using the CellTiter-Glo assay. (C) Flow cytometric analysis of Annexin V‑FITC/PI‑stained SKOV-3 cells treated with the control (0.1% DMSO) or 0–100 μM DFMO was used to determine the apoptotic rates of SKOV-3 cells under various concentrations of DFMO. (D) Caspase-3 activity in DFMO-treated SKOV-3 cells was measured using the Luciferase Assay (Caspase-Glo 3/7 Assay system). (E) Western blot and quantification analysis for the apoptotic proteins Bcl‑2, Bcl-xL, Bax, cleaved caspase‑3, and cleaved PARP in cells treated with DFMO. The controls were treated with 0.1% DMSO. Data are expressed as mean ± SD. * P < 0.05, ** P < 0.01, *** P < 0.001 compared with the controls.

Figure 3. Effect of combination therapy using DFMO and cisplatin on SKOV-3 cell survival and apoptotic cell death. Cell viability was determined of cells treated with (A) 10 μM cisplatin, 100 μM DFMO or (B) 10 μM cisplatin/0–100 μM DFMO for 48 h using the PrestoBlue assay and CellTiter-Glo assay. (C) proliferation was determined of cells treated with (C) 10 μM cisplatin, 100 μM DFMO, or (D) 10 μM cisplatin/0–100 μM DFMO for 48 h. (E) Flow cytometric analysis of Annexin V‑FITC/PI‑stained SKOV-3 cells treated with 10 μM cisplatin/0–100 μM DFMO for 48 h was performed to determine the apoptotic rates of SKOV-3 cells under various concentrations of DFMO. (F) Caspase-3 activity in cisplatin/DFMO-treated SKOV-3 cells was measured using the Luciferase Assay (Caspase-Glo 3/7 Assay system). (G) Western blot and quantification analysis for the apoptotic proteins Bcl‑2, Bcl-xL, Bax, cleaved caspase‑3, and cleaved PARP in cells treated with DFMO. The controls were treated with 0.1% DMSO. Data are expressed as mean ± SD. *P < 0.05, **P < 0.01, ***P < 0.001 compared with the controls.

Figure 4. Induction of AP-1 activity and JNK signaling using the combination therapy of DFMO and cisplatin. An AP-1 luciferase plasmid vector was transiently transfected into SKOV-3 cells for 48 h. The transfected cells were seeded in cell culture plates and treated with cisplatin or DFMO for 48 h. The AP-1 luciferase activity of DFMO- or cisplatin/DFMO-treated cells was measured using the Luciferase Assay System (A, D). Western blot and quantification the proteins of JNK signaling (phospho-JNK, JNK, phospho-cJun, AP-1, c-Fos) and apoptosis (phospho-Bad, Bad) in DFMO- or cisplatin/DFMO-treated cells (B, C, E, F). The controls were treated with 0.1% DMSO. Data are expressed as mean ± SD. *P < 0.05, **P < 0.01, ***P < 0.001 compared with the controls.

Figure 5. Induction of DFMO-induced apoptotic cell death through the regulation of AP-1 activity. The transfected cells were seeded in cell culture plates and treated with DFMO, SR11302, or DFMO/SR11302 for 48 h. (A) The AP-1 luciferase activity in the treated cells was determined using the Luciferase Assay System. (B) Western blot of organelle extracts (10 μg) was performed using the following markers: nucleus; HDAC, Histone H3/ cytosol: Hsp90, GAPDH. (C) Western blot analysis of the JNK signaling pathway detected JNK signaling proteins (phospho-JNK, JNK, phospho-cJun, AP-1), anti-apoptotic proteins (Bcl-2, Bcl-xL), and pro-apoptotic proteins (Bad, Bax cleaved caspase-3, cleaved PARP) in DFMO-, SR11302-, or DFMO/SR11302-treated cells, respectively. The controls were treated with 0.1% DMSO. Data are expressed as mean ± SD. *P < 0.05, **P < 0.01, ***P < 0.001 compared with the controls.

Reviewer 2 Report

All my comments were successfully implemented. I recommend acceptance in the present form. Authors my refrain from further ICC staining of Annexin and Caspase-3

Author Response

Thank you very much for the review of our manuscript (IJMS-1368450). The comments of the review were constructive and have been used to revise and improve the manuscript. We enclose a revised version of our paper titled “Difluoromethylornithine Induces Apoptosis through Regula-tion of AP-1 Signaling via JNK Phosphorylation in Epithelial Ovarian Cancer” by Woo Yeon Hwang, Wook Ha Park, Dong Hoon Suh, Kidong Kim, Yong Beom Kim, Jae Hong No as a submission to International Journal of Molecular Science. We marked up with the “Track changes” function to make our revisions. The following is an itemized account of the changes in the manuscript made in response to the comments.

Point 1: All my comments were successfully implemented. I recommend acceptance in the present form. Authors my refrain from further ICC staining of Annexin and Caspase-3

Response 1: Thank you very much for your acceptance.

Round 3

Reviewer 1 Report

All comment have been addressed. One important point, the author only provide the quantification analysis of Fig-1E, 3G, 4B and 5C, but not show that P-value

Author Response

Response to Reviewer 1 Comments

Thank you very much again for the review of our manuscript (IJMS-1368450). The comments of the review were constructive and have been used to revise and improve the manuscript. We enclose a revised version of our paper titled “Difluoromethylornithine Induces Apoptosis through Regula-tion of AP-1 Signaling via JNK Phosphorylation in Epithelial Ovarian Cancer” by Woo Yeon Hwang, Wook Ha Park, Dong Hoon Suh, Kidong Kim, Yong Beom Kim, Jae Hong No as a submission to International Journal of Molecular Science. We marked up with the “Track changes” function to make our revisions. The following is an itemized account of the changes in the manuscript made in response to the comments.

Point 1: All comment have been addressed. One important point, the author only provide the quantification analysis of Fig-1E, 3G, 4B and 5C, but not show that P-value

Response 1: Thank you for providing valuable comment. We provided P-value of quantification analysis of Fig-1E, 3G, 4B and 5C.

Modified text:

Figure 1. Inhibition of cell survival and induction of apoptotic cell death by DFMO in SKOV-3 cells. (A) SKOV-3 cells were cultured with varying concentrations of DFMO (from 0 to 100 μM) for 48 h, and cell viability was detected using the PrestoBlue assay. (B) Cell proliferation was determined by analyzing ATP content in DFMO-treated SKOV-3 cells using the CellTiter-Glo assay. (C) Flow cytometric analysis of Annexin V‑FITC/PI‑stained SKOV-3 cells treated with the control (0.1% DMSO) or 0–100 μM DFMO was used to determine the apoptotic rates of SKOV-3 cells under various concentrations of DFMO. (D) Caspase-3 activity in DFMO-treated SKOV-3 cells was measured using the Luciferase Assay (Caspase-Glo 3/7 Assay system). (E) Western blot and quantification analysis for the apoptotic proteins Bcl‑2, Bcl-xL, Bax, cleaved caspase‑3, and cleaved PARP in cells treated with DFMO. The controls were treated with 0.1% DMSO. Data are expressed as mean ± SD. * P < 0.05, ** P < 0.01, *** P < 0.001 compared with the controls.

Figure 3. Effect of combination therapy using DFMO and cisplatin on SKOV-3 cell survival and apoptotic cell death. Cell viability was determined of cells treated with (A) 10 μM cisplatin, 100 μM DFMO or (B) 10 μM cisplatin/0–100 μM DFMO for 48 h using the PrestoBlue assay and CellTiter-Glo assay. (C) proliferation was determined of cells treated with (C) 10 μM cisplatin, 100 μM DFMO, or (D) 10 μM cisplatin/0–100 μM DFMO for 48 h. (E) Flow cytometric analysis of Annexin V‑FITC/PI‑stained SKOV-3 cells treated with 10 μM cisplatin/0–100 μM DFMO for 48 h was performed to determine the apoptotic rates of SKOV-3 cells under various concentrations of DFMO. (F) Caspase-3 activity in cisplatin/DFMO-treated SKOV-3 cells was measured using the Luciferase Assay (Caspase-Glo 3/7 Assay system). (G) Western blot and quantification analysis for the apoptotic proteins Bcl‑2, Bcl-xL, Bax, cleaved caspase‑3, and cleaved PARP in cells treated with DFMO. The controls were treated with 0.1% DMSO. Data are expressed as mean ± SD. *P < 0.05, **P < 0.01, ***P < 0.001 compared with the controls.

Figure 4. Induction of AP-1 activity and JNK signaling using the combination therapy of DFMO and cisplatin. An AP-1 luciferase plasmid vector was transiently transfected into SKOV-3 cells for 48 h. The transfected cells were seeded in cell culture plates and treated with cisplatin or DFMO for 48 h. The AP-1 luciferase activity of DFMO- or cisplatin/DFMO-treated cells was measured using the Luciferase Assay System (A, D). Western blot and quantification the proteins of JNK signaling (phospho-JNK, JNK, phospho-cJun, AP-1, c-Fos) and apoptosis (phospho-Bad, Bad) in DFMO- or cisplatin/DFMO-treated cells (B, C, E, F). The controls were treated with 0.1% DMSO. Data are expressed as mean ± SD. *P < 0.05, **P < 0.01, ***P < 0.001 compared with the controls.

Figure 5. Induction of DFMO-induced apoptotic cell death through the regulation of AP-1 activity. The transfected cells were seeded in cell culture plates and treated with DFMO, SR11302, or DFMO/SR11302 for 48 h. (A) The AP-1 luciferase activity in the treated cells was determined using the Luciferase Assay System. (B) Western blot of organelle extracts (10 μg) was performed using the following markers: nucleus; HDAC, Histone H3/ cytosol: Hsp90, GAPDH. (C) Western blot analysis of the JNK signaling pathway detected JNK signaling proteins (phospho-JNK, JNK, phospho-cJun, AP-1), anti-apoptotic proteins (Bcl-2, Bcl-xL), and pro-apoptotic proteins (Bad, Bax cleaved caspase-3, cleaved PARP) in DFMO-, SR11302-, or DFMO/SR11302-treated cells, respectively. The controls were treated with 0.1% DMSO. Data are expressed as mean ± SD. *P < 0.05, **P < 0.01, ***P < 0.001 compared with the controls.
